# Energy optimization induces predictive-coding properties in a multi-compartment spiking neural network model

**Mingfang Zhang[1¤], Raluca Chitic[2], Sander M. Bohté** [ID][1,3]*

1 Centrum Wiskunde & Informatica, Amsterdam, The Netherlands, 2 University of Groningen, Groningen, The Netherlands, 3 Swammerdam Institute for Life Sciences (SILS), University of Amsterdam, Amsterdam, The Netherlands

¤ Current address: ENS-PSL, Ecole normale supérieure, Paris, France
* S.M.Bohte@cwi.nl

**Data availability statement:** All data and code used for running experiments, model fitting,

## Abstract

Predictive coding is a prominent theoretical framework for understanding hierarchical sensory processing in the brain, yet how it could be implemented in networks of cortical neurons is still unclear. While most existing studies have taken a hand-wiring approach to creating microcircuits that match experimental results, recent work in rate-based artificial neural networks revealed that suitable cortical connectivity might result from self-organisation given some fundamental computational principle, such as energy efficiency. As no corresponding approach has studied this in more plausible networks of spiking neurons, we here investigate whether predictive coding properties in a multi-compartment spiking neural network can emerge from energy optimisation. We find that a model trained with an energy objective in addition to a task-relevant objective is able to reconstruct internal representations given top-down expectation signals alone. Additionally, neurons in the energy-optimised model show differential responses to expected versus unexpected stimuli, qualitatively similar to experimental evidence for predictive coding. These findings indicate that predictive-coding-like behaviour might be an emergent property of energy optimisation, providing a new perspective on how predictive coding could be achieved in the cortex.

## Author summary

Predictive coding is an elegant and influential theoretical framework for understanding learning and processing in the brain, with several experimental findings seemingly in support. Yet, current predictive coding frameworks require specific connectivity motifs to be implemented whose emergence so far has remained unexplained – instantiated with

and plotting is available on a GitHub repository at https://github.com/sbohte/SNN_PC_Multicomp. We have also used Zenodo to assign a DOI to the repository: 10.5281/zenodo.13329388.

**Funding:** SB is supported by NWO NWA ORC grant NWA.1292.19.298 and the European Union (grant agreement 7202070 "HBP"). The funders had no role in study design, data collection and analysis, decision to publish, or preparation of the manuscript.

**Competing interests:** The authors have declared that no competing interests exist.

spiking neurons, such motifs become even more intricate and more difficult to explain. An alternative point of view assumes that the brain is capable of efficient deep learning in some manner, for example energy optimization in rate-based RNNs can result in network behavior reminiscent of predictive coding. However, real biological networks differ from RNNs in important ways: first, they operate in continuous time rather than sequential steps, and second, real biological neurons emit binary spikes, which for instance makes it difficult to communicate an error that could be positive or negative. Defining an internal energy-measure for multi-compartment spiking neurons, we demonstrate how the resulting recurrent networks can exhibit several predictive coding like-properties when optimizing for both task and energy efficiency. The energy optimized network then demonstrates lower overall activity, generative behavior, and differential responses to expected vs unexpected stimuli. Energy-minimization in multi-compartment spiking neurons can thus bring tangible benefits and explain predictive-coding like empirical findings.

## 1. Introduction

Predictive coding is a prominent theory of sensory processing in the brain, postulating that the brain learns a generative model of the world capable of predicting sensory inputs through hierarchically organized brain areas [1,2]. Although indirect experimental evidence and computational models built with predictive coding principles have successfully explained various experimental phenomena, the precise neural implementation of predictive coding remains a subject of debate [3,4]. Proposed algorithms disagree in terms of the neuronal types and connectivities, with disparate views for what cortical microcircuits are involved in the implementation of predictive coding [3,5–8].

Computational models of microcircuits are instrumental in uncovering computational mechanisms and generating novel hypotheses for understanding predictive coding in the cortex. However, existing proposals of predictive coding are subject to two main constraints that limit their accuracy in capturing the corresponding cortical mechanisms. First, most predictive coding algorithms involve specific wiring between distinct neuronal types with particular abstractions that might be ill-founded. For instance, the classical formulations of predictive coding implement separate error and prediction neurons within each layer or area (Fig 1A), though limited evidence supports functionally distinct sub-populations [3,9,10]. Some studies apply a highly constrained one-to-one correspondence between error and prediction neurons within individual cortical regions (Fig 1A), which is a biologically problematic assumption about the cortex [11,12]. Another example is the specific wiring between excitatory and inhibitory neurons to create three functionally different groups for encoding representations, positive, and negative errors in the canonical circuit proposed by [3] (Fig 1B). While these models are built to implement predictive coding principles, it is difficult to argue that these specifically hardwired microcircuits are precisely present in the cortex. The second major limitation is that most models are non-spiking networks which lack biological realism [9,13–15]. This has been mostly due to the lack of a straightforward way to transfer classical rate-based predictive coding to a spiking implementation, ie. spiking neurons cannot signal negative errors without specific wiring. The difficulties in training spiking neural networks have also hindered efforts in this direction [4]. Additional trade-offs occur between biological fidelity and scalability, which makes it difficult to study more complex phenomena in a biological network [9,11,14,15]. The few studies implementing predictive coding in

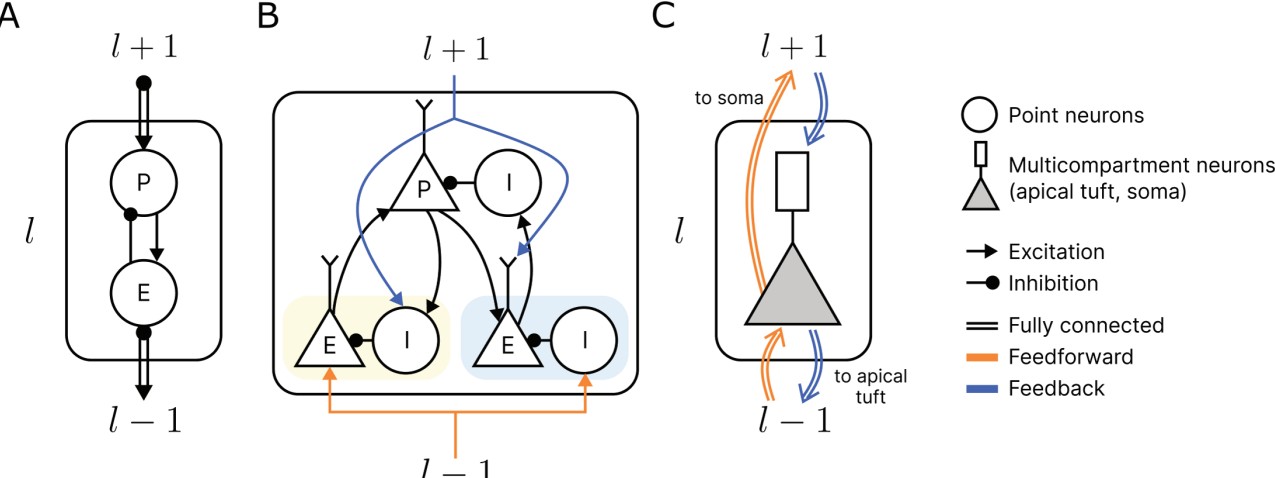

**Fig 1. Example microcircuits of predictive coding.** (A) Classical predictive coding from [12] with separate error (*E*) and prediction (*P*) neurons in each layer. A one-to-one connection is imposed between error and prediction populations. (B) Schematic proposal from [3] with specific wiring between excitatory and inhibitory neurons to encode positive errors (*E* yellow background), negative errors (*E* blue background), and representations/predictions (*P*). Connectivity types between neuron populations were uncategorized. (C) Our architecture with multicompartment neurons in each layer. The rectangle denotes the apical tuft compartment and the triangle denotes the somatic compartment. Fully connected feedforward signals are integrated at the soma (triangle) and feedback signals at the apical tuft (rectangle). In contrast to the hard-wiring approach in (A) and (B), our proposal does not assume the presence of specialised neuron types or circuits.

spiking neural networks like in [11] leave a gap in the literature for more biologically realistic network models without specific architectural biases.

To build a model without these limitations, we take inspiration from several approaches adopting a gradient optimisation approach to investigate the relationship between more fundamental computational principles and structural-functional properties [16–18]. We argue that if the network exhibits cortical properties after being optimised for a particular objective, it means that that objective could also be optimised in the brain and be a driving force in the learning of cortical connectivities. This approach allows us to hypothesize and test the more fundamental computational goals that give rise to neural properties. Ali et al. [16] demonstrated the potential of this approach by showing that energy optimisation in a rate-based recurrent artificial neural network led to the prediction of input at the next time step via inhibition, aligning with the classical formulation of predictive coding. The authors argued that predictive coding can result from self-organisation as the cortex optimises for energy efficiency, extending the connection between predictive coding and energy efficiency [19–23]. Moreover, their findings suggest that predictive coding microcircuits do not have to be hard-wired in the cortex, but can instead be an emergent attribute of a system with some fundamental architectural components in place. Although the conclusions from [16] have limited generalizability due to the use of a single-layer non-spiking network, they provide a new perspective on predictive coding implementation in the cortex based on gradient optimisation.

In this work, we apply the optimisation approach in more detailed and biologically plausible spiking neural networks. Inspired by recent progress on predictive coding in spiking and multi-compartment neurons [8,11,24], in particular the somato-dendritic error mismatch scheme proposed by [25], we create a multi-layer multi-compartment spiking neural network that can be trained in a supervised fashion using gradient optimisation. The question we ask is

whether energy optimisation would induce predictive-coding-like behaviour. Following the core notion in [25], we define the energy loss as a function of the voltages in the separate compartments of each spiking neuron in the model. We hypothesise that within a multi-layer network with basic feedforward and feedback connections between areas, an additional 'internal' energy loss optimised alongside a task loss will be enough for predictive-coding-like behaviour to emerge. After training, we evaluate two unique properties supporting predictive coding: the models' capabilities of reconstructing internal representations with top-down expectation signals and their differential responses to expected versus unexpected stimuli. We find that the energy-optimised network is capable of holding internal representations of expected stimuli in the absence of actual input, similar to what was found in the human brain [26,27]. We also qualitatively replicate the empirical results showing differential responses in both apical tuft and somatic voltage of neurons when perceiving expected versus unexpected stimuli [28]. The unique presence of these properties in the energy-optimised model demonstrates that when optimizing for an energy minimization objective, predictive-coding-like behaviour can be learned without pre-specified connectivity. Additional analyses find that network training results in stable internal connectivity despite the possibility of spiking saturation due to positive feedback loops. Overall, this work demonstrates that using an optimisation approach in spiking neural networks can inform the underlying computational principles driving the emergence of predictive coding circuits and produce models that match experimental results.

## 2. Methods

### 2.1. Neuron and network model

Taking inspiration from previous approaches [8,29–31], we construct a simple multi-compartment spiking neuron model that mimics a pyramidal cell in the cortex (Fig 1C). Each neuron has two compartments: a dendritic compartment representing the apical tuft of a neuron and a somatic compartment. The apical tuft integrates inputs from higher areas in the hierarchically organised network, while the soma directly integrates feed-forward information [32–35]. Voltage in the apical tuft unidirectionally affects the soma potential. As we focus on object classification in visual hierarchical processing, which involves mainly the inter-layer interactions, we omitted the details of basal dendritic sites to arrive at a simple neuronal model where bottom-up inputs are directly integrated into the soma [8,24]. This setup captures some key aspects of the current understanding of cortical connectivity patterns between areas [36].

The neuron model's spiking mechanism is modelled as in the Adaptive Leaky-Integrate-and-Fire (ALIF) model, a LIF neuron augmented with an adaptive firing threshold [37]. The spiking of a neuron $i$ is a function of the somatic membrane potential $V_{s,i}(t)$ and the spiking threshold $b_i(t)$: spikes $\{t_j\}$ from a neuron $j$ are modeled as Dirac delta-functions $\delta_j^{l+1}(t - t_j)$, and a neuron $i$ emits a spike if the somatic membrane potential at time $t_i$ exceeds the threshold from below, emitting a spike $\delta_i^l(t - t_i)$. Three factors, the voltage at the apical tuft ($V_{a,i}(t)$), the somatic membrane potential ($V_{s,i}(t)$), and the adaptive threshold ($b_i(t)$), affect the spiking dynamics of each neuron (Fig 2A). Adopting the idiom of the deep learning field, we refer to the calculation of neural activity patterns in the network given inputs as "inference". At each time step of inference, each neuron simultaneously traces the top-down and bottom-up signals in the somatic and apical compartments respectively. The apical dendritic compartment receives top-down spike-trains from the next layer and its voltage evolves according to:

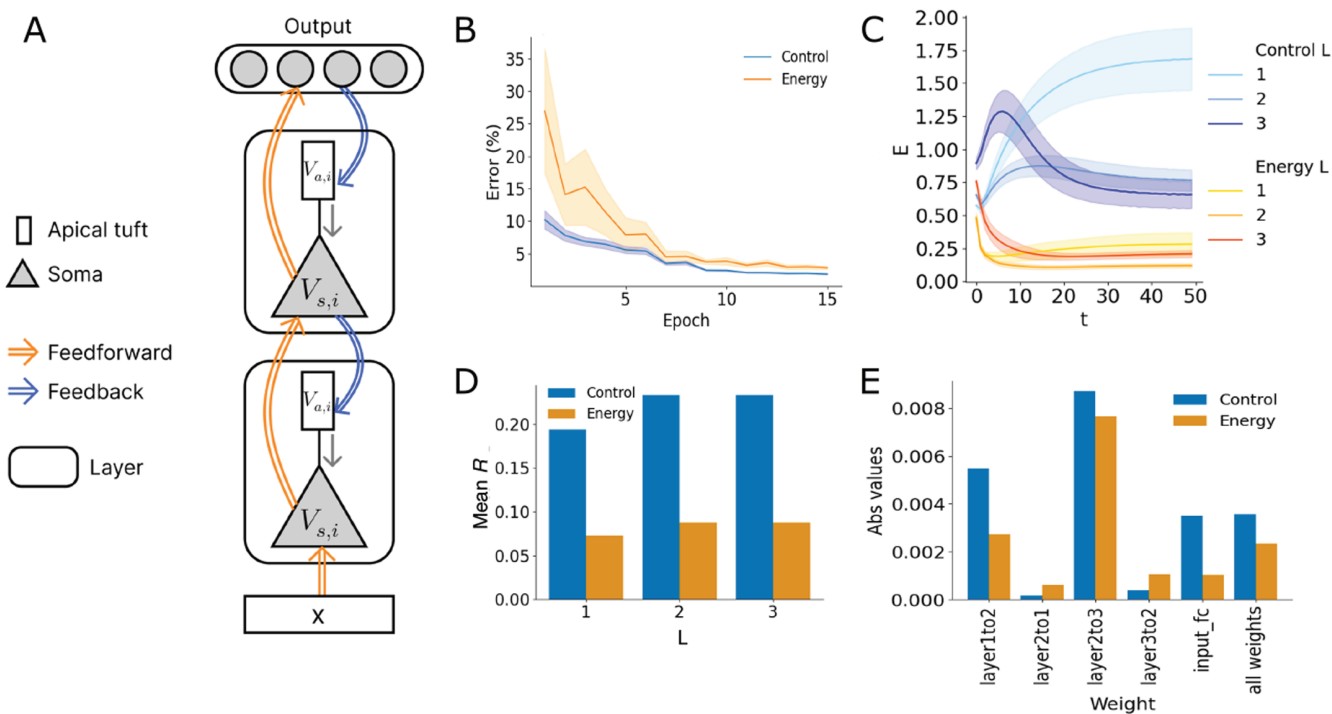

**Fig 2. The energy model optimises for energy compared to control.** (A) Schematic illustration of network architecture with multi-compartment spiking neurons. Only two layers are shown here. Feedforward connections project to the somatic compartment of neurons in the next layer while feedback connections project to the apical tuft dendrite compartment of the previous layer. The voltage at the apical tuft uni-directionally affects the somatic membrane potential. The output neurons are non-spiking membrane potential integrators that determine the predicted class. (B) Test error per epoch for both models. Results from ten models, initialised with different random seeds, for each condition are assessed. (C) Energy per neuron averaged over all samples as the mean absolute voltage difference between soma and apical tuft compartments. All layers in the energy model show lower energy than the control model. (D) Mean spike rate per layer in the energy and control models. Neurons in the energy model spike less across all layers. (E) Absolute values of feedforward and feedback weights. The left panel plots each set of weights separately. The right panel plots all weights and shows that the energy optimisation also results in smaller weights in the energy model. Error bars in all sub-figures plot 95% confidence interval.

$$\frac{dV_{a,i}^l}{dt} = -\frac{V_{a,i}^l}{\tau_{a,i}} + \sum_j W_{ij}^{FB} \sum_{\{t_j\}} \delta_j^{l+1}(t), \tag{1}$$

where $V_{a,i}^l$ is the apical voltage of the $i$th neuron in layer $l$, $\tau_{a,i}$ is the time constant for the apical site, and $W_{ij}^{FB}$ is the feedback weights from layer $l + 1$ to $l$. The membrane potential at the somatic compartment evolves following

$$\frac{dV_{s,i}^l}{dt} = -\frac{V_{s,i}^l}{\tau_s} + \sum_j W_{ij}^{FF} \sum_{\{t_j\}} \delta_j^{l-1}(t - t_j) + f\left(V_{a,i}^l(t)\right) - b_i^l(t) \sum_{\{t_i\}} \delta_i^l(t - t_i), \tag{2}$$

where the $V_{s,i}^l$ is the somatic membrane potential of the $i$th neuron in layer $l$, $\tau_{s,i}$ is the corresponding time constant, and $b_i^l(t)$ is the adapted spiking threshold at time $t$. The somatic compartment voltage is directly influenced by the feed-forward signal, in the form of spikes from the previous layer $S_j^{l-1}(t)$ weighted by feed-forward weights ($W_{ij}^{FF}$) from layer $l–1$ to $l$, and the voltage at the apical tuft. We let the strength at which the voltage from the apical tuft

$(V_{a,i}(t))$ drives the soma be determined by a shifted sigmoid function $f(x)$, defined as:

$$f(x) = \frac{1}{4} \tanh(x/2).$$
(3)

Inspired by [17], this function bounds the influence from apical tuft at each time step for both positive and negative voltage ranges. The unidirectional influence from the apical tuft to the soma means that over time only the somatic compartment integrates two sources of input from the lower and higher hierarchical areas.

Whether an ALIF neuron spikes at a given time step is additionally dependent on the adaptive spiking threshold $b_i(t)$, which is determined by:

$$b_i(t) = b_0 + \beta\eta_i(t),$$
(4)

where $b_0$ is the baseline threshold, $\eta_i(t)$ is the adaptive contribution term, and $\beta$ is a constant (default value 1.8) that determines the size of adaptation of the threshold. The adaptive contribution to the spiking threshold of each neuron evolves following:

$$\frac{d\eta_i^l}{dt} = -\frac{\eta_i^l}{\tau_{adp,i}} + \sum_{\{t_i\}} \delta_i^l(t - t_i),$$
(5)

where $\tau_{adp,i}$ is the time constant that determines the decay rate of $\eta_i^l$. Whenever a neuron receives sufficient bottom-up and top-down inputs such that a spike is emitted, the increase in $\eta_i^l(t)$ raises the spiking threshold, making the neuron less likely to spike again at the next time step. After spiking, the somatic potential undergoes a soft reset through a spike-triggered refractory response of size $b_i^l(t)$ (Eq. 2), retaining the amount in the potential that exceeds the threshold due to the time step effect. Overall, the spiking dynamics of each neuron in the network are determined by a combination of feed-forward and feedback inputs, an adaptive spiking threshold, and the time constants that control the decay rates of each dynamical variable developing in the neuron.

The studied network architecture is composed of three layers of multi-compartment spiking neuron models (L1, L2, L3) (Fig 2A). In each layer, the neurons receive spiking input from both the lower and higher layers via fully connected weights, where a bias is implemented for each neuron as a constant current injection through a trainable weight, $W_{ib} \cdot 1$, as in [42]. The output layer is comprised of non-spiking leaky neurons that integrate inputs through membrane potentials following

$$\frac{dV_{i,mem}}{dt} = \frac{-V_{i,mem}}{\tau_{mem}} + \sum_j W_{ij}^{FF} \sum_{\{t_j\}} \delta_j^{l-1}(t - t_j),$$
(6)

where $V_{i,mem}$ is the membrane potential of one output neuron, $\tau_{mem}$ is the time constant, and $\{\delta_j^{l-1}(t_j)\}$ is the spike-train from L3. Due to the non-spiking nature of these output neurons, we first L2-normalise their membrane potentials before passing them as directly injected currents through the feedback weights from the output to layer 3 (Fig 2A). Overall, the network can be seen as a fully connected network with feedforward and feedback connections with internal recurrence within the dynamics of each neuron. Inputs are injected at each timestep as a constant spike-train proportional to the intensity of the input value, as in [38]. Training adjusts all weights $W$ as well as all decay-constants $\tau$, the latter specific for each

individual neuron. The Eqs (1), (2), (5), and (6) are calculated using the Euler forward method, see S1 Text.

## 2.2. Training and task

We implement supervised training of the networks, both with and without an energy-loss term, to investigate whether predictive coding properties can arise due to energy optimisation. The training process utilizes a combination of the online learning algorithm Forward Propagation Through Time (FPTT) and surrogate gradients, which enables end-to-end optimisation using gradient descent within the Pytorch auto-differentiation framework [38–40,42]. The Forward-Propagation-Through-Time (FPTT) algorithm [40], which enables training of complex spiking neural networks on classification tasks [38], allows updates of parameters at each or every K timesteps (K-step updates) during the sequence. We apply K-step=10 updates during training as we find that empirically yielded the best results. Unlike the more standard Backpropagation Through Time (BPTT) algorithm, where parameters are updated once at the end of each sequence, FPTT achieves online learning through immediate updates to network parameters by optimizing a dynamic regularizer in addition to the task-relevant loss [40]. As we show in our results, FPTT resulted in better learning of feedback weights in the energy models than classical Backpropagation Through Time (BPTT). For the surrogate gradient, we apply the Multi-Gaussian surrogate gradient introduced in [42] which was shown to consistently outperform other surrogate gradients.

At each update step during training, the parameters are optimised with respect to a global loss, which contains a task-relevant loss and the dynamic FPTT regularizer. In the energy optimization condition, an energy term is added to the global loss function as an additional regularizer to be optimised. Within our multi-compartment neuronal model, we defined an energy term $\mathcal{L}_{E,t}$ using a function $g$ of the apical tuft and soma compartments voltages at the time of update:

$$g\left(V_{a,i}^l(t), V_{s,i}^l(t)\right) = |V_{a,i}^l(t) - V_{s,i}^l(t)|,$$
$$\mathcal{L}_{E,t} = \left(\sum_l \sum_i g\left(V_{a,i}^l(t), V_{s,i}^l(t)\right)\right)/N, \tag{7}$$

where $g()$ computes the absolute difference between the voltages and $\mathcal{L}_{E,t}$ is the average of all outputs of $g$ in the network ($N$: total number of neurons). Here, the voltages from different compartments are used to compute the membrane potential that determines the spiking dynamics. In [25], such a difference signal is computed via a dedicated inter-neuron projecting the somatic output to the apical tuft. Our version is thus a roughly equivalent efficient implementation. Alternatively, biological neurons may compute this separate signal $g()$ via some biochemical pathway in the neuron diffusing from soma to apical tuft. The signal $g()$ can be interpreted either as the electric potential energy local to each neuron, or alternatively be regarded as a comparison between the integrated feedforward and feedback signals within each neuron. The overall loss optimised during training at each learning step follows:

$$\mathcal{L}_t = \mathcal{L}_{clf,t} + \alpha_{reg}\mathcal{L}_{reg,t} + \alpha_E\mathcal{L}_{E,t}, \tag{8}$$

where $\mathcal{L}_{clf,t}$ is the task-related classification loss (Negative Log-Likelihood), $\mathcal{L}_{reg}$ is the dynamic FPTT regularizer, and $\alpha_E$, $\alpha_{reg}$ are constant scalars for weighting respective regularizers. An energy-optimised model was trained with $\alpha_E = 5 \cdot 10^{-2}$ and the control model with $\alpha_E = 0$. We use the AdamX optimiser [43] and apply dropout as well as weight decay during the training to reduce overfitting.

We train the network to perform MNIST handwritten digit classification. The MNIST dataset consists of 60,000 training and 10,000 test samples which were normalised during preprocessing. The network runs inference for T time steps on each image and is reinitialised between samples. The log softmax values of output membrane potentials determine the predicted class. At the beginning of inference for each batch of samples, spiking neurons are initialised with somatic membrane potentials uniformly distributed between 0 and 1. All $\eta_i$ and $V_{a,i}$ are set to 0 and $b_0$ to 0.1 at the beginning of each inference (see Tables 1 and 2 for all hyperparameter settings). Network weights were initialised with Xavier initialisation [44] and all bias terms were initialised to 0 prior to training, ie $W_{ib}$ = 0. Hyperparameters are determined with reference to [42].

## 3. Results

### 3.1. The energy model shows lower inter-compartmental and spiking energy than the control

We initialise ten models for each condition with different random seeds to assess model performance. After training, models of both conditions achieve good performance on the MNIST classification test set, with an error rate of $1.83(\pm0.07, 95\%ci; 98.17\%$ accuracy) for the control models and $2.41(\pm0.07)$, (97.59% accuracy) for the energy models (Fig 2B). We find that this decrease in accuracy for the energy-networks is gradual as a function of the strength of energy-regularization, that is, larger $\alpha$ setting result in fewer spikes and lower accuracy. This finding in consistent with literature [41] and we selected a value for $\alpha$ that still resulted in high accuracy.

One energy-optimised model and one control model trained with FPTT are randomly selected for the subsequent analyses, where the control model was studied at equal accuracy as

**Table 1. Training hyperparameters.**

| Hyperparameter | Value |
|---|---|
| Epoch | 10 |
| Learning rate | $1e{-}3$ |
| Decay rate | $1e{-}4$ |
| $\alpha_E$ | $5e{-}2$ or 0 |
| $\alpha_{reg}$ | 1 |
| K step | 10 |
| Drop out | 0.4 |
| T (steps) | 50 |
| Layer sizes (spiking) | 600,500,500 |
| Output layer size | 10 |
| $dt$ (ms) | 0.5 |

**Table 2. Initialisation values for hyperparameters of each neuron. All time constants were initialised to have normal distributions centred around the values presented in the table with a standard deviation of 0.1. All output neurons had τmem initialised to be the same constant.**

| Hyperparameter | Value |
|---|---|
| $\tau_s$ (ms) | 15 |
| $\tau_a$ (ms) | 15 |
| $\tau_{adp}$ (ms) | 20 |
| $\tau_{mem}$ (ms) | 5 |
| $b_0$ | 0.1 |

the energy model by selecting an accuracy-matching earlier check-point – all findings also held up when using the fully trained control model.

We first validate that the energy model indeed consumes less energy than the control model (Fig 2). We assess this using two key metrics: energy computed by $g\left(V_{a,i}^l(t), V_{s,i}^l(t)\right)$ per neuron across samples, and the average spike rate of each layer per sample. During inference on the test set, which we run for $T$ time steps, the mean energy for each layer in the energy model is lower than their counterparts in the control model and consistently stabilises at a value below the initial level, indicating the additional energy loss successfully induced energy optimisation in the network (Fig 2C). We then compute the mean spike rate per layer in response to each sample in both models: we find that the energy-optimized model emitted fewer spikes compared to the control model (Fig 2D). This could be attributed to the significantly lower mean absolute weights of the feedforward connections, akin to synaptic transmissions, which resulted in smaller contributions to the energy consumption of the energy model overall (Fig 2E). We also see here that overall the energy model has smaller weights than the control model. Having adaptive thresholds was critical, as removing adaptation by setting $\beta = 0$ in (4) resulted in poor accuracy and significantly increased firing rates, both for the control model and in particular for the energy model (Supporting Information S4 Fig). The trained time-constants of the neurons do not contribute to the differences in energy consumption as the distributions are similar across both models (Supporting Information, S1 Fig). These findings demonstrate that by minimizing the inter-compartmental voltage difference, a measure of the electrical potential energy within each neuron, we concurrently achieve reduced spiking and synaptic transmission, which are two main sources of neuronal energy consumption [45]. In particular, this also establishes the voltage difference as a valid proxy for energy consumption in these models.

## 3.2. Only the energy model can reconstruct internal representations given top-down signals

We next ask whether the energy-trained model can generate internal representations with occluded or no inputs. Not only is the brain able to imagine visual objects, experiments have also shown that the retinotopic areas where visual input is occluded within a larger image contain information about the image, which could be explained by the activation of those areas due to top-down projections carrying predictions or context given the non-occluded parts of the visual stimuli [26]. We conduct a similar experiment on the trained networks to see whether we could replicate this result. To decode from the spiking representations, we first train a linear decoder to reconstruct the test sample from the spiking pattern (vector containing the average spikes per neuron across inference time) in a particular layer (Fig 3A). The decoder is trained to minimise MSE loss between the projected image and the actual test sample via gradient descent (using the Adam optimiser) over 20 epochs. The error curves of decoder training for both models (Fig 3B) demonstrate that the linear decoder successfully converged when fitting to the training data. One decoder is trained for each layer from each model and used to decode what information the internal representations of the networks contain.

We first test the networks with a half-occluded image randomly sampled from a class (eg. number 3 in Fig 3C) with the correct class clamping in the output layer to mimic top-down predictive projections from processing areas downstream to the visual cortex. During clamping, the membrane potentials of the output layer are fixed to be the same vector throughout inference on one sample, where the membrane potential of the output neuron for the intended class was set to 1 and others were set to -1, which modelled perceiving a partially

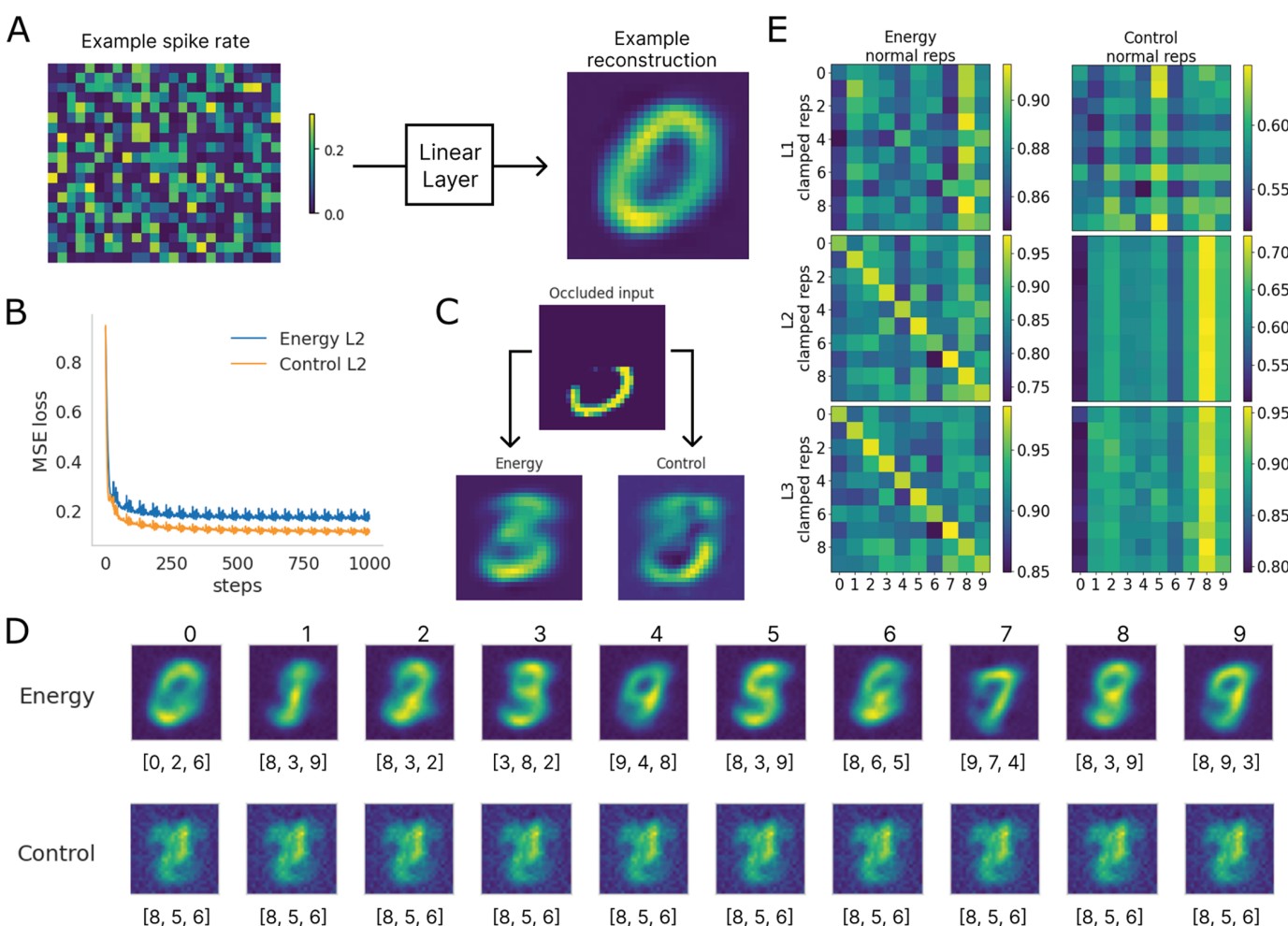

**Fig 3. Reconstructive capacity of the energy model.** (A) Illustration of the decoding setup. A linear decoder is trained to reconstruct the original image from the spike rate representation of one layer along $T$ steps. An example heat map of the spike rate over $T$ time steps is shown on the left. On the right is an example projected image from the spiking representation. (B) MSE loss during training of decoders for L2. Both decoders were able to fit the training set well. (C) Comparison of network inference with correct class clamping. The energy model can fully reconstruct the digit while the control model does not perform meaningful reconstruction of internal representations. (D) Decoded internal representations with class clamping absent input. Below each decoded digit, the top 3 classification as obtained from the respective models given these digits as input. Only the energy model reconstructs class-specific internal representations. Clamped representations from the control model are indistinguishable between classes. (E) Pair-wise representational similarity of clamped vs normal representations in the energy and control models. A clear class-specific representational structure is present in the energy model while absent in the control model.

occluded image with internal expectations of the image class. In both the occluded and no input conditions, models are given $5T$ steps for inference to compensate for the reduced inputs and to leave sufficient time for top-down projections to take effect. As shown in Fig 3C, with the correct class clamping, the energy model's internal representation from the L2 is able to fill in the occluded parts while the control model does not perform meaningful reconstruction. Presenting the correct class clamping induces the energy-optimised network to reconstruct the intended image '3'. Notably, if a uniformly distributed noise vector is used to clamp the output neurons, the energy model reconstructs different digits in the internal representations with repeated sampling of noise (Fig 4). This demonstrates that internal representations in the energy model differed depending on the prior when the input was ambiguous.

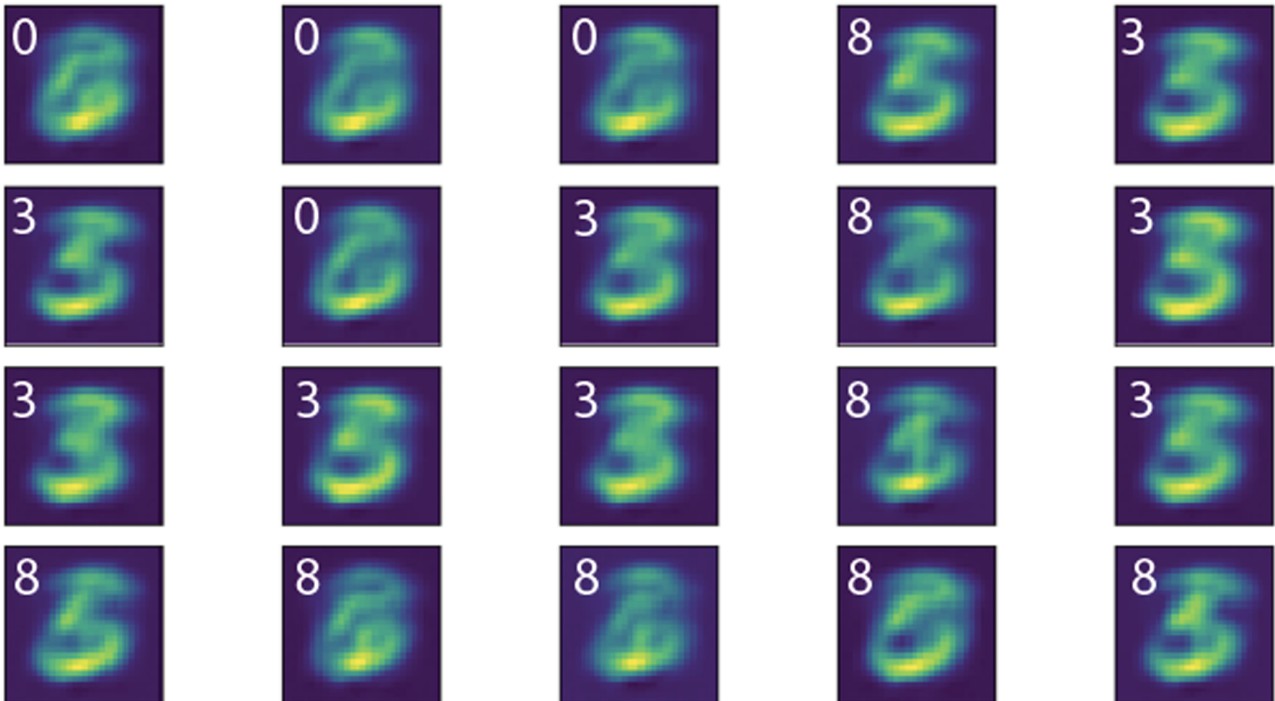

**Fig 4. Decoded clamped representation with ambiguous occluded input and noise vector clamping.** The same occluded sample '3' from Fig 3C and the same decoding scheme were used to decode representations from L3 of the energy model. A vector with uniform noise was used to clamp the output neurons during inference. The figure shows samples generated from 20 samples of random noise. The decoded images show that depending on the noisy prior, the energy model would internally represent different digit classes (eg. '0', '3', '8', inset number denotes the digit the sample is classified as when presented as input.).

We next test the models' capabilities to reconstruct without any input (pixel values equal to 0) and with only clamping. The same clamping and decoding methods are used as described above for internal representations from the models over $5T$ time steps of inference with class clamping. We find that only the energy model's spiking representations could be decoded into digits while those in the control model are indistinguishable between classes (Fig 3D). This is further verified by a Representation Similarity Analysis (RSA) [46] of the per-class representations of networks in the normal inference condition (with input) or the clamped condition (no input) (Fig 3E). We compute the normal representations for each class by averaging the spike rate patterns for each layer over all samples of each class. The clamped representations are taken as the spike rate pattern per layer given a clamped class. The pair-wise similarities were computed as 1 minus the cosine distance of normal and clamped representations per class. As shown in Fig 3E, the clamped representations in the energy model show a clear class-specific structure, where the clamped representation is most similar to the normal representation from the corresponding class; this pattern was not observed in the control model. After grouping pair-wise similarities into same-class or different-class similarities across layers, results further confirmed that the clamped representation in the control model does not contain any class-specific information (Supporting Information S2 Fig). All these results indicate that only the energy model has the capability of reconstructing, thus predicting the inputs when top-down signals are provided as a prior for disambiguating or imagining the inputs. The energy regularizer induced effective learning of

feedback weights such that representations in a higher layer could spatially predict the bottom-up signals received by the lower layer.

### 3.3. Neurons in the energy model respond differentially to expected vs unexpected stimuli

One neural phenomenon at the foundation of predictive coding is that neurons respond differentially to expected versus unexpected stimuli [28,47]. We thus ask whether the energy model would exhibit such properties. To evaluate this, we designed a match/mismatch experiment that simulates scenarios of expected versus unexpected stimuli for the trained networks, thereby probing if the neuronal response within the energy model varied across these conditions (see Fig 5A and Fig 5B). In this experiment, the models initially receive no stimuli. Upon stimulus onset, an image sample is introduced as normal, accompanied by clamping at the output neurons. This clamping corresponds to either the actual class of the image (representing the match or expected condition) or an incorrect class (representing the mismatch or unexpected condition) (Fig 5A). The membrane potential of the output neuron linked to the clamped class is subsequently set to 1, while those of all other neurons were set to -1: by clamping the top-down information in the network, we create an environment wherein the information relayed from higher hierarchical areas of the brain either corroborated or contradicted the bottom-up input. The presentation of the stimulus and the associated clamping is followed by an additional phase of zero input, marking the conclusion of the inference process (Fig 5B). Both the class of the presented stimulus and the clamped class are randomly selected.

To compare the extent of differential response to expected and unexpected stimuli in the energy and control models, we compute a Mean Signed Difference (MSD) in voltage signals between match and mismatch conditions in each compartment ($V_{a,i}^l(t)$, $V_{s,i}^l(t)$) for each neuron within one layer during stimulus presentation:

$$\sum_{t=T/4,i=1}^{t=3T/4,i=|l|} \frac{Vma_{c,i}^l(t) - Vmm_{c,i}^l(t)}{T/2 \times |l|} \tag{9}$$

where $|l|$ denote the size of layer $l$, and $Vma$ and $Vmm$ denote the potential (either apical or somatic) in the match and mismatch conditions respectively.

Comparing the distribution of these MSDs in each compartment in the energy and control model, we observe that significantly more neurons in L2 of the energy model have larger MSD between conditions in voltage traces than in the control model, indicating the energy model exhibits a greater differential response to unexpected stimuli (Fig 5C and Fig 5D). Example voltage traces of neurons with different MSD values are presented in Fig 5G, where we observe diverging voltage values during stimulus presentation in a subset of neurons. This is also reflected in the differences in spike rates ($\delta R$) per neuron during stimulus presentation between conditions in different models (Fig 5E). Kurskal Wallis tests on the distributions of MSD in voltage traces and $\delta R$ all yielded significant differences (Supporting Information S1 Table). We proceed to compute the $\delta R$ per neuron across all three layers (Fig 5F). This reveals that L3 in the energy model - the highest in the processing hierarchy - displays a markedly more pronounced divergence in spiking responses between conditions relative to the lower layers. This could be attributed to the hierarchical nature of our model, wherein the upper layers are primarily driven by top-down signals, while lower layers are chiefly influenced

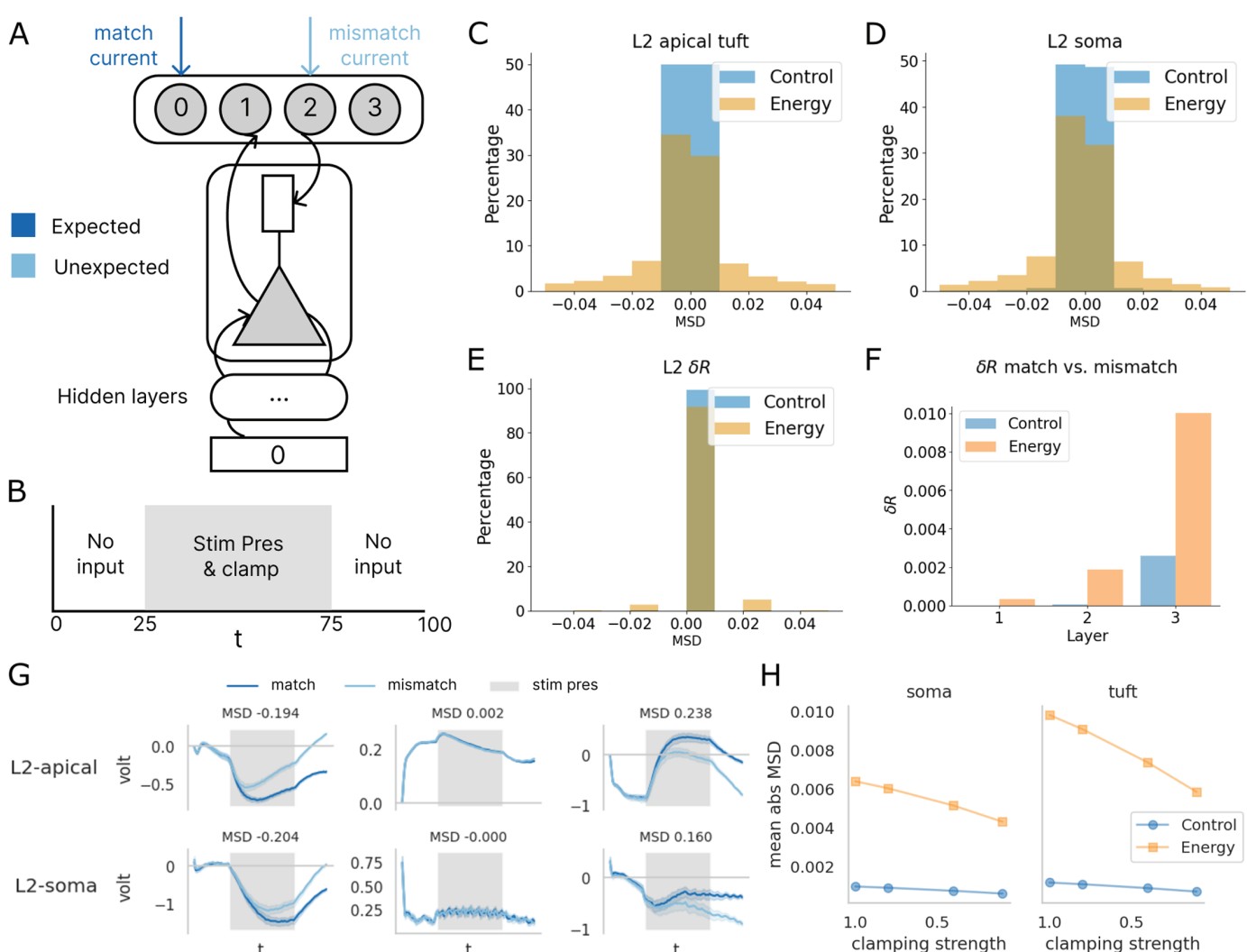

**Fig 5. The energy model responds more differentially to unexpected stimuli than the control model.** (A, B) Illustration of the match/mismatch experiment. Either the correct or wrong class of output neuron is clamped at the output layer. The models are given $T$ time steps for inference, with no inputs on either end of stimulus presentation ("stim pres"). (C, D, E) Distributions of apical tuft compartment voltage difference, soma voltage difference, and spike rate difference between conditions. All differences are computed for the time steps during stimulus presentation ($t = 25 - 75$). Across all three metrics, the energy model shows a more drastic response difference between expected and unexpected stimuli than the control model (Supporting Information S1 Table). (F) Difference in spike rate between experimental conditions across layers. (G) Examples of voltage trajectories in the apical tuft and soma compartments from single neurons from layer 2 in the energy model with different MSD. (H) MSD responses decrease more strongly for the energy model compared to the control model, as a function of decreased top-down clamping, for both somatic and tuft response.

by inputs [33]. We remark that this suggests a novel prediction that can be validated through neural recordings from different cortical areas in match/mismatch experimental paradigms.

The MSD measure also enables us to predict the effect of a purported increase in classification uncertainty: we decrease the amount of top-down clamping of during stimulus representation, while equally distributed this decrease to increased activation of the other classes. As shown in Fig 5H, the result of this manipulation is a smooth and substantial decrease of the mismatch difference in the energy model, while there is barely a change in the control model.

In all, these findings show that the energy model successfully replicated the experimental outcomes delineated in [28] while making specific predictions for novel manipulations like classification uncertainty. At the same time, such properties were notably absent in the control model. We thus infer that the observed predictive coding properties of the network, the distinct response to expected and unexpected stimuli, can be attributed to the energy optimization in the energy model.

### 3.4. The internal connectivity is stable in both models

We next ask whether the trained networks are stable. Given that the energy model is optimised for matching bottom-up and top-down projections, a neuron might end up receiving both excitatory feed-forward and excitatory feedback inputs in a potentially positive feedback loop, leading to over-excitation that would result in instability in firing (eg. saturation of spiking, Fig 6A). To confirm the stability of the models, we vary the amount of current input into the networks by adding or subtracting pixel values from the preprocessed images. Subtracting pixel values increases inputs as currents into the neurons due to negative weights associated with the negative pixel values (Supporting Information S3 Fig). We find that overall the spike rates of neurons in both models responded roughly linearly to variations in the average current input into the neurons, that is, the larger the positive currents, the higher the resulting firing rates (Fig 6B). This is not due to saturation of spiking in the neuron, as most neurons respond in a graded fashion to increase in spike-rates: in Fig 6C, we plot for 20 randomly selected neurons in L1 the slope for the change in firing rate as a function of modulation of the input. We applied 5 different levels of input modulation and measures the change in firing rate for each level. We used linear regression to fit this change to a linear response model, obtaining excellent fits with $r^2 > 0.98$ for all neurons; Fig 6C plots the slope for each of the 20 neurons. The energy model's response also varies less than the control model for the same amount of input manipulation due to smaller input weights in the network, which could also explain the lower performance deterioration in test accuracy with different input intensities (Supporting

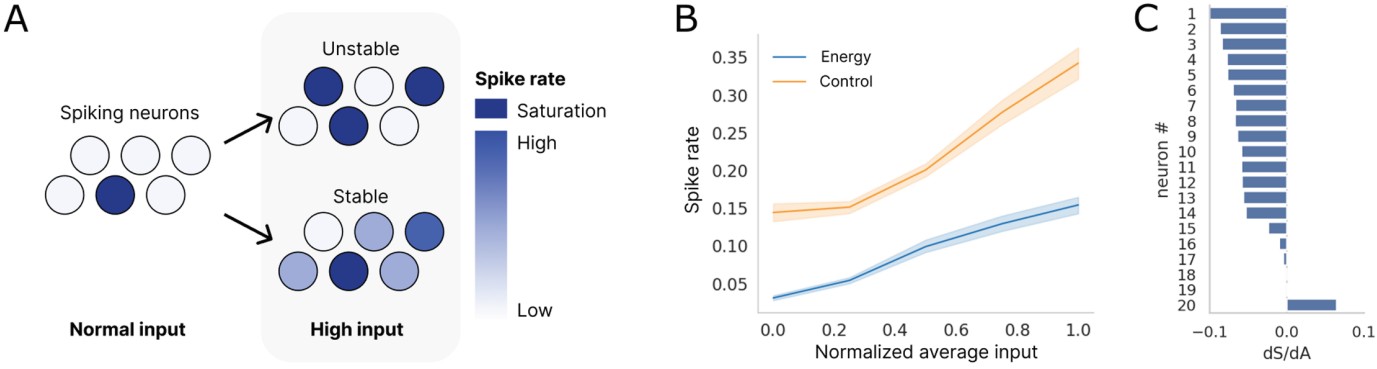

**Fig 6. Model stability.** (A) Illustration of stable and unstable networks. The spiking neurons in the network emit spikes at respective spike rates given inputs at baseline intensity. In a stable network, most neurons respond with a higher spike rate with stronger input, resulting in a higher overall spike rate in the network. In an unstable network, the few neurons that are linked in positive feedback loops via feedforward and feedback weights either barely spike or show saturated spiking at every time step with stronger input. The colour of neurons in the illustration indicates their spike rate. Neurons with the darkest blue have saturated spiking. (B) The average spike rate of all neurons in the model per sample in relation to normalised average input into each neuron in L1. (C) Slope for the spike rate as a function of input modulation (dS/dA) for 20 individually sampled neurons in L1 of the energy model. Most sampled neurons show a graded response in spike rate to input pixel manipulation.

Information, (Supporting Information S3 Fig). Overall, these results demonstrate that the trained models were indeed stable and reflect the intensity of inputs through spike rates.

## 3.5. FPTT results in more effective learning of feedback weights in the energy model than BPTT

Finally, we investigate whether the distinct temporal credit assignment mechanisms of FPTT and BPTT would lead to any substantial differences in the properties of trained networks. We train an additional energy model using BPTT and contrast its reconstructive capabilities with the FPTT-trained energy model. To quantify the reconstructive quality, we computed the cosine distance between the decoded images from spiking representations and the mean pixel values of images from each class. We find that the BPTT-trained energy model can internally represent different digit classes with no input and only top-down clamping, yet the quality of reconstruction from each layer is consistently lower than that from the FPTT-trained energy model (Fig 7B and Fig 7C). The reconstruction quality is particularly worse in L1 of the BPTT-trained energy model, indicating degradation of temporal credit assignment in BPTT towards lower areas in the network processing hierarchy. The class structure in the clamped representations of the BPTT-trained energy model is also less pronounced (Fig 7D) than in the FPTT-trained energy model. Overall, given that reconstruction relies mainly on feedback projections between adjacent layers, these results demonstrate the less effective learning of feedback weights in the BPTT-trained energy model. This suggests that the temporal locality

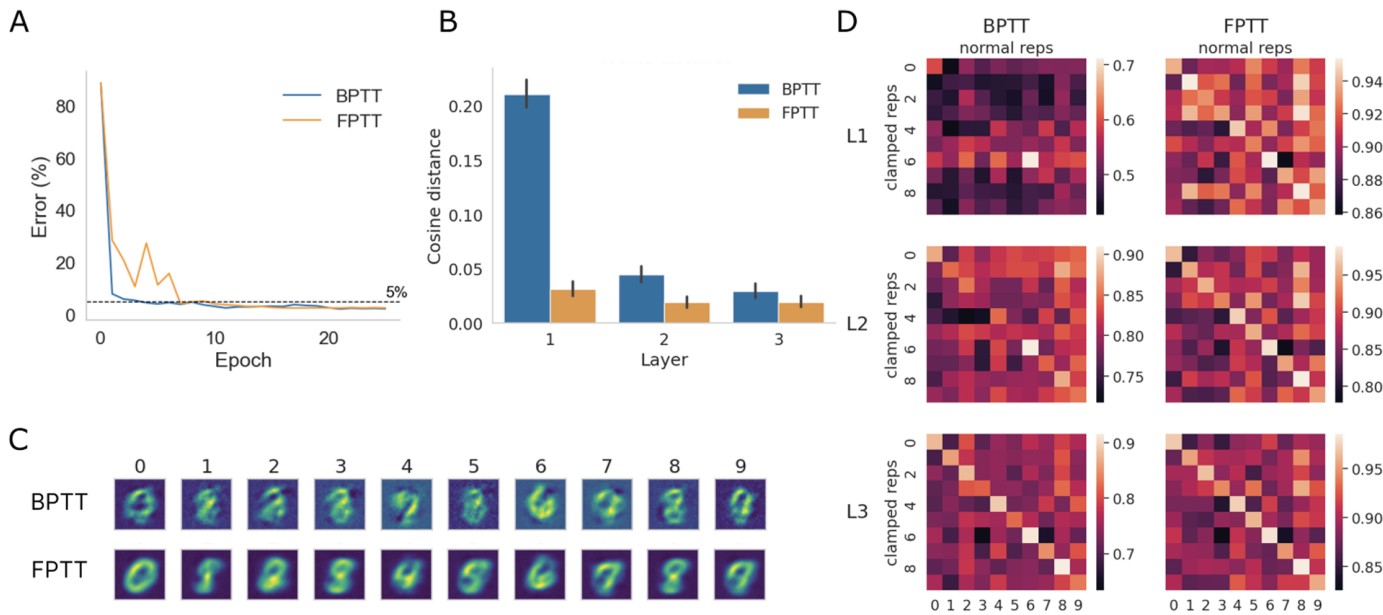

**Fig 7. Reconstructive capacities of BPTT- vs FPTT-trained energy models.** (A) Test errors from BPTT-trained and FPTT-trained energy models. The error rate is 2.19% for the BPTT model and 2.49% for the FPTT model. (B) Quality of reconstructed internal representations in BPTT- vs FPTT-trained energy models. The cosine distances are computed between the mean pixel values of each class and decoded images from internal clamped spiking representations from respective models. Across layers, the clamped class representations in the BPTT-trained energy model are more different from the class mean image. (C) Decoded images from internal spiking representations in L2 of BPTT-trained and FPTT-trained energy models. The quality of decoded images from the BPTT-trained energy model is lower than those of the FPTT-trained energy model (FPTT decoded images repeated from Fig 3D for comparison). (D) Pair-wise representational similarity of clamped vs normal representations in BPTT-trained and FPTT-trained energy models. The class-specific representational structure in the energy model is less pronounced than that in the FPTT-trained energy model.

of credit assignment might be crucial in credit assignment for feedback weights in a hierarchically organised network.

## 4. Discussion

We demonstrate that energy optimisation, formulated as a function of voltages in different compartments within each neuron in a spiking neural network, is a potential computational principle for driving cortical learning to produce predictive-coding properties. Our energy loss, which can be interpreted as the electrical potential energy within each neuron, proves to be an adequate proxy for energy consumption as it encourages reduced spiking and synaptic weights in the energy model. Given a main classification objective, the spiking network trained with an additional energy loss is able to learn feedback weights that predict the bottom-up inputs. The resulting energy model also replicates experimental findings supporting the theory of predictive coding, such as the reconstruction of inputs with only top-down feedback [26] and differential responses to expected versus unexpected stimuli [28], where our model additionally predicts that increasing classification uncertainty is associated with decreased differential responses. These predictive coding properties are observed only in the energy model, thus providing support for the hypothesis of energy efficiency underlying the emergence of predictive-coding-like behaviour in the cortex. The weights are successfully trained to produce stable internal connectivity in the network despite the potential for positive feedback loops in the network. Additionally, our results indicate that feedback weights in the FPTT-trained energy model are learned more effectively than those in a corresponding BPTT-trained energy model, demonstrating the benefit of choosing a learning algorithm with temporal locality in its credit assignment.

Our results yield two notable implications. First, our network communicates predictions between layers, as opposed to errors. Since the matching of bottom-up and top-down signals is carried out implicitly within individual neurons, this dispensed the necessity for discrete error and prediction neurons within each area. Second, the feedback weights essentially serve to reverse the feature extraction operations performed by the feed-forward weights. This aligns with the hierarchical predictive coding scheme, where abstract representations from higher cortical areas must be translated into more sample-specific representations within earlier cortical areas, acting as a form of prediction. In our network configuration, this translation is accomplished directly by the feedback weights linking prediction neurons between layers, rather than linking prediction to error neurons. While our network model does not mirror the specific laminar organisation of the cortex, it offers an alternative perspective for the transference of information between cortical areas. This direct mapping from higher to lower hierarchical representations may elucidate how non-stimulated cortical areas retain information about the visual context and mental imagery [26,27,48].

Our formulation of energy loss is closely related to other studies on energy and neural computation, such as [25]. We design the energy loss to be the absolute difference between the somatic and apical tuft compartment, which essentially represents the difference between bottom-up and top-down signals received by a single neuron. By minimising this term that captures the electric potential energy within each neuron, the network learns to match representations across layers and also optimises energy consumption both in terms of spiking and synaptic transmission. While this is different to some other approaches (eg. the Free Energy Principle [2] which centres around thermodynamic energy), many types of energy are involved and interchangeable during the metabolic processes of a neuron. Including energy minimization between top-down and bottom-up representations also implies that the network optimizes the somewhat conflicting objectives of learning top-down class-generic

representations and bottom-up sample-specific representations. This conflicting objective is the likely cause of accuracy decreasing as a function of strength of the energy loss. Our empirical results on the overall reduction in spiking and synaptic transmission in the energy model call for a more in-depth mathematical analysis into how our formulation of energy could be related to other ones.

Our results diverge from the findings in [16] which asked a similar question of whether energy optimisation gives rise to predictive coding using a different setup and using classical rate-based artificial neurons. Ali et al [16] optimised the pre-activation of ReLU units as energy in a one-layer recurrent network inferencing on predictable sequences. As a result, they found that units in the recurrent layer self-organised into separate prediction and error neurons and that prediction occurred as within-layer inhibition to counter the excitatory inputs. This is different from our results which showed that top-down predictions were present as excitatory signals, with feedback weights creating a direct mapping of predictions from higher to lower layers.

The disparities between these findings and conclusions predominantly stem from differences in the conceptual frameworks, setups of the network model and tasks, as well as the chosen definitions of energy loss. First, [16] studied unsupervised temporal prediction in a discrete-time rate-coded recurrent network, a problem that becomes fundamentally different in spiking neural networks operating over continuous time. While the rate-coded network just switches to a new prediction at every discrete time step, the spiking neural network would have to hold the current prediction over a time interval until it's time to switch, turning temporal sequence prediction into a nontrivial decision-making problem (maintain vs switch) at every moment in continuous time (see e.g. [49] for modelling of continuous decision-making in basal ganglia for action-selection). This fundamental difference between the networks makes it difficult to make a direct meaningful comparison between the findings of the two studies. Combining learning with decision-making could however be a way to study dynamic datasets in future work. Second, we are interested in using multiple processing layers to model visual hierarchical processing, yet [16] was focused on self-organisation within one recurrent layer in a temporal prediction problem. The distinct energy definitions are also more meaningful in their respective network contexts. In these separate setups, the findings regarding whether the prediction signals should be excitatory or inhibitory are optimal for each system: within-layer inhibitory recurrent drive minimises preactivation as energy, and top-down excitatory projection that matches bottom-up input minimises intercompartment voltage difference as energy. It is possible that these processes coexist in the cortex, just as [8] argued that dedicated error neurons could exist together with the dendritic implementation of predictive coding. While the visual ventral pathway could employ the mechanisms in our model to link abstract and sample-specific representations along the visual areas, cortical areas responsible for sequence learning could have inhibitory temporal predictions as shown in [16], implemented with additional mechanisms to solve the problem in continuous time. Prospective investigations might consider using our multi-compartment, multi-layer spiking network configuration for a temporal prediction task analogous to that in [16] to ascertain whether congruent or different outcomes are achieved. Empirical evidence regarding the existence of specialized error neurons within specific sensory processing pathways would ultimately help determine which model offers a better mechanistic explanation for various types of sensory processing in the cortex.

Our work was inspired by the recent studies of dendritic predictive coding yet different in one subtle way. Existing proposals of dendritic error computation implement algorithms such that the error value is explicitly encoded by the voltage of the apical dendritic compartment

and used to guide local voltage-dependent plasticity rules [8,24,29]. In [24], this was achieved by wiring up specific interneurons to dendritic sites of pyramidal neurons. There is some experimental evidence supporting the involvement of inhibitory interneurons producing predictive error and even gating the plasticity of feedforward synapses [50–52]. In our model, the error value is represented implicitly, computed as the difference in voltage between compartments in our model. A natural question is thus how the neurons can utilise this internal value for learning at the synapse. The empirical literature that directly examines this phenomenon is relatively sparse, though [53] presented some experimental evidence supporting the presence of implicit error information within each neuron. One possibility is that since the membrane potential, which determines the neuronal spiking, is a non-linear summation between the voltages of two compartments, biological neurons could compute another signal with these voltages as a representation of their internal electric potential energy to drive synaptic plasticity. Our energy loss, which calculates the absolute difference in voltages between distinct compartments in spiking neurons, models this computation that could potentially be carried out by specific biochemical pathways. This energy loss, unlike a simple spike-counting loss, may be critical for the presence of predictive coding features in the energy model because its implicit information about the mismatch between feedforward and feedback signals could be the driving force behind the learning of top-down weights for predictions. Our model thus offers a new perspective on the potential relevance of this internal energy term in synaptic plasticity. Alternatively, as noted earlier, we can consider our multi-compartment neuron as an abstraction of a small circuit where the somatic output is signalled to the apical tuft using a dedicated interneuron, similar to the proposal in [25].

This current work, which involved a simple classification task using an internal energy loss, can be extended in several ways to test the generalizability of its framework and conclusions. To start with, we chose supervised learning to model top-down projections from multimodal-associative areas downstream to the visual ventral stream as a form of supervision signals in the brain [54–56]. However, several recent studies have shown that networks trained unsupervised or self-supervised have representations that better correlate with brain representations and better predict human perception and behaviour than supervised networks [57–59]. Therefore, it would be interesting to explore the optimisation of energy in an unsupervised or self-supervised training scheme. A larger dataset with more naturalistic images could also be used to test more complex network properties. Another possibility is implementing different architectures for different tasks. For instance, we have not included within-layer lateral recurrent connections that are important for visual recognition [60–62]. We also omitted local lateral inhibition which has been shown to play a role in plasticity for memory and learning [63]. Future work could thus extend energy optimisation work with in-layer recurrent neural networks for a temporal task. To study spatial organization for instance in thalamocortical circuits [64], and compare to known biological observations, the fully connected architecture could be replaced by more natural connectivity patterns mimicking receptive fields. In terms of the learning algorithm, we chose FPTT, which is a temporally local but spatially global algorithm. It would be interesting to explore whether other online algorithms could replicate these results. Similarly, the surrogate gradient used in the backward pass is principally sensitive to spike-rates; learning rules that are more sensitive to spike-times could be investigated to study the potential for spike-time-based top-down prediction canceling predictable bottom-up inputs [65]. Future work could also incorporate more complex dendritic computations or implement Dale's law in the network to examine the resulting self-organisation due to energy efficiency [66,67].

Our present study demonstrates that predictive coding properties in a multicompartment spiking neural network may arise from the optimization of each neuron's internal energy. Empirically, we have connected this energy loss to a decrease in synaptic transmission and spiking, proposing an optimization technique that produced models capable of replicating experimental findings. This approach paves the way for further exploration of the link between energy optimization and predictive coding in spiking neural networks.

## Supporting information

**S1 Text. Euler approximations to dynamical systems.**
(PDF)

**S1 Table. Kurskal Wallis Test statistics for distributions in Fig 2C, Fig 2D and Fig 2E.**
(PDF)

**S1 Fig. Densities of trained time constants in both models**. The energy and control models have similar time constants after training.
(TIF)

**S2 Fig. Same vs different class representation similarity between clamped and normal representations.** We aim to evaluate whether clamped representations exhibited greater similarity to normal representations from the corresponding class as opposed to those from different classes. To achieve this, we group pairwise representation similarities into two categories: same-class similarities are representation similarities between normal and clamped representations from the same class, while different-class similarities are those between different classes. In the energy model, the same-class similarities are significantly higher than different-class similarities. In the control model, there is no significant difference between similarity types, indicating a lack of class information in the clamped representations of the control model.
(TIF)

**S3 Fig. Model Reaction to Modifications in Pixel Values**. Left: Accuracy of models tested on manipulated images. The x-axis shows the changes in pixel values introduced to the preprocessed test set images. The control model's test accuracy shows a steeper decline as pixel values strayed from the standard range. This effect may be attributed to more significant alterations in the input currents to L1 neurons at each level of pixel manipulation in the control model (Right). The control model's comparatively larger input weights potentially account for this observed trend (Refer to Fig 2E).
(TIF)

**S4 Fig. Non-adapting spiking neurons.** Removing adaptation from the spiking neuron model decreases accuracy for both models (Left), in particular for the energy model and increases the average firing rates in all layer (Right).
(TIF)

## Acknowledgments

The authors express their appreciation to Bojian Yin for his insightful recommendations concerning SNN training.

## Author contributions

**Conceptualization:** Mingfang Zhang, Raluca Chitic, Sander Bohte.

**Data curation:** Mingfang Zhang.

**Formal analysis:** Mingfang Zhang, Raluca Chitic, Sander Bohte.

**Funding acquisition:** Sander Bohte.

**Investigation:** Mingfang Zhang.

**Methodology:** Mingfang Zhang, Raluca Chitic, Sander Bohte.

**Software:** Mingfang Zhang, Raluca Chitic.

**Supervision:** Sander Bohte.

**Validation:** Sander Bohte.

**Visualization:** Mingfang Zhang, Sander Bohte.

**Writing – original draft:** Mingfang Zhang.

**Writing – review & editing:** Mingfang Zhang, Raluca Chitic, Sander Bohte.

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
