## [Decision Letter · Decision Letter 0]

15 Oct 2024

Dear Prof Dr. Bohte,

Thank you very much for submitting your manuscript "Energy Optimization Induces Predictive-coding Properties in a Multicompartment Spiking Neural Network Model" for consideration at PLOS Computational Biology.

As with all papers reviewed by the journal, your manuscript was reviewed by members of the editorial board and by several independent reviewers. In light of the reviews (below this email), we would like to invite the resubmission of a significantly-revised version that takes into account the reviewers' comments.

The reviewers praise your study and do not challenge the novelty of the results or the validity of your model to provide new insights. They appreciate the code quality and availability and constructively suggest specific clarifications and relatively minor corrections. However, reviewer #1 also raises some questions about the model breadth and proposes a series of tests to gauge it.

We cannot make any decision about publication until we have seen the revised manuscript and your response to the reviewers' comments. Your revised manuscript is also likely to be sent to reviewers for further evaluation.

1 A letter containing a detailed list of your responses to the review comments and a description of the changes you have made in the manuscript. Please note while forming your response, if your article is accepted, you may have the opportunity to make the peer review history publicly available. The record will include editor decision letters (with reviews) and your responses to reviewer comments. If eligible, we will contact you to opt in or out.

2 Two versions of the revised manuscript: one with either highlights or tracked changes denoting where the text has been changed; the other a clean version (uploaded as the manuscript file).

Sincerely,

Emili Balaguer-Ballester, PhD

Academic Editor

PLOS Computational Biology

Daniele Marinazzo

Section Editor

PLOS Computational Biology

Dear authors,

The reviewers praise your study and do not challenge the novelty of the results or the validity of your model to provide new insights. They appreciate the code quality and availability and constructively suggest specific clarifications and relatively minor corrections.

However, reviewer #1 also raises some questions about the model breadth and proposes a series of tests to gauge it.

Yours sincerely,

Emili Balaguer-Ballester, PhD.

Reviewer's Responses to Questions

**Comments to the Authors:**

Reviewer #1: The authors propose a multi-compartment spiking neural network model, trained with energy-based optimization, and show that this induces some aspects of predictive coding properties that are consistent with observations in biological neural networks, in particular primary visual cortex. The paper is well written, with a detailed introduction and problem statement, and a thorough description of the methods. I appreciated the availability of the source code, which I was able to study in detail. The results are encouraging, but the paper would have greater impact if the results were generalised beyond what is currently presented in the paper, justifying a major revision.

A major novelty is the introduction of energy, described in eq (7), which justifies the use of two compartment models. However, some results do not seem surprising (e.g. the caption of Figure 2 reads "The energy model optimises for energy compared to control.") and "Neurons in the energy model respond differentially to expected vs unexpected stimuli". Have you checked whether these effects are progressive as a function of \alpha_E, or whether they appear after a certain threshold? As in other papers (such as doi:10.1162/neco_a_01325), the addition of this code inevitably reduces accuracy (as observed in the lower error rates in the control), so other measures of the benefits of predictive coding should be put forward. Stability to contrast levels or noise is a good place to start, but you could also investigate the effect of feedback on the neural response, for example by looking at whether the presentation of a sequence of digits would differentially encode different inducers of the same digit (e.g. an instance of "3" followed by another of "3") or different digits ("3" - "7"). You can probably guess that the effect would be similar to that observed in mismatch negativity. Going further, one could actually test a prediction by titrating the uncertainty of a guess by clamping the output to a single hot output modulated by a contrast, such as (.05, .05, .05, .05, .55, .05, .05, .05, .05, .05, .05, .05, .05) and observing the activity to a novel stimulus.

Minor points

--------------

First, the model is quite complex and uses elements from previous work by the same authors. While FPTT is compared with BPTT, the effect of the parameters of ALIF (especially by turning off adaptation) was not tested and could be a nice addition.

Secondly, the clarity of Figure 6c could be improved as it is difficult to relate the bars to each other. You seem to want to show that the curves are less saturated - so perhaps you could do fits with saturation functions (e.g. naka-Rushton curves = 1 / (1 + (x/x_50)**-n) ) and plot the histogram of slopes n

Also, what would be the guessed numbers in Fig. 4 from these reconstructions?

Fig. S.4. should be made more quantitative by plotting some sort of "tuning curve" of this neuron to different bars and see if the control shows the same behaviour.

Finally, I would have many questions regarding using other datasets than a static one (MNIST), yet it seems perhaps going beyong the scope of this paper.

Minor comments

-------------------

P1 Check and simplify the syntax of this long sentence "Here, we demonstrate how recurrent networks comprised of multi-compartment ..."

p2 "that when when optimizing" > "that when optimizing"

p5 Eq (3) is equivalent to 1 / 4 . tanh(2 . x) which is commonly used in other models. or maybe simply tanh as weights would scale activity ?

Fig 2. (d) misses labels on both axis. complete "figures plot 95%ci." > "figures plot 95% confidence intervals."

Fig3 could be improved by showing bigger images + it could be useful to show the guess given by the inference for these generated images

Fig 6 : the axis labels are not written in proper English, and the legend should be made more explicit (reading the paper makes it obvious what E vs nE are but readers may get confused "Energy-based" and "Control" could be more informative). y-labels in (c) are not necessary "(b) Spike rates of 20 individually sampled neurons in L1 of the energy" > "(c) Spike rates of 20 individually sampled neurons in L1 of the energy"

Reviewer #2: In their manuscript "Energy Optimization Induces Predictive-coding Properties in a Multicompartment Spiking Neural Network Model", Zhang et al. investigate whether and how adding an additional loss term related to a notion of consumed energy, when training a spiking neural network with gradient descent, leads to the emergence of properties otherwise associated with predictive coding ideas. They find that these properties arise and only do so if the "energy loss" is included.

The manuscript is overall very well written and the results appear to be backed up by the numerical investigation. The figures are informative and well-presented.

The code underlying the work is publicly available on Github, allowing detailed inspection and reproduction of the work.

While the manuscript is overall easy to understand there are some minor issues with the description of the methods that need to be fixed before publication and some minor writing issues:

Methods:

1. The description of the neural network model is inconsistent and mathematically incorrect. The authors define S_i(t) as taking values 0 or 1, the latter when a neuron's voltage reaches the firing threshold, at which time the neuron is instantaneously reset to 0 by subtracting the threshold. Subsequently S_i(t) is used on the right hand side of ODEs for the voltages. However, with this definition, they have no effect on the voltage (the set of times S_i(t) is non-zero is a null set and finite value 1 hence has no influence). I believe the authors are mixing a continuous time description in their methods section with the discrete time implementation in their code to arrive at this inconsistent/incorrect description.

2. On page 5, fully connected weights (with bias): What does "with bias" mean? Please define unambiguously by a formula.

3. :"After spiking, the somatic potential undergoes a soft reset of the current value of b^l_i(t) (Eq. 2), retaining the amount in the potential that exceeds the threshold due to the time step effect." -> everything is described in continuous time, so this statement makes no sense (see 1 above, same issue of mixed descriptions).

4. The Methods section does no elaborate how the presented ODEs are solved numerically.

5. It would be helpful to already mention explicitly in the methods that timescales (tau_x) are individual to neurons and also subject to training

6. On page 7, formatting of "5e-2" should be in maths notation

7. "At the beginning of inference for each batch of samples, spiking neurons are initialised with somatic membrane potentials uniformly distributed between 0 and 1 at the beginning of training." -> broken sentence

8. "all bias terms were initialised to 0 prior to training." -> as above, please explain explicitly how biases work in your SNN model

In the results:

1. The classification error rates achieved appear to differ in a statistically significant way - please add commentary on this in the paper.

2. Fig 5 caption: "on either end of stimulus onset.": The stimulus onset is the point in time when the stimulus starts; I believe the authors meant "stimulus presentation". This confusion repeats several times throughout the paper

3. Please define the Mean Signed Difference by an explicit equation

4. "response also varies less than the control model in for the same amount" -> broken sentence

**Have the authors made all data and (if applicable) computational code underlying the findings in their manuscript fully available?**

Reviewer #1: Yes

Reviewer #2: Yes

PLOS authors have the option to publish the peer review history of their article (what does this mean?). If published, this will include your full peer review and any attached files.

Reviewer #1: **Yes: **Laurent U Perrinet

Reviewer #2: **Yes: **Thomas Nowotny
---

## [Decision Letter · Decision Letter 1]

28 Jan 2025

PCOMPBIOL-D-24-01385R1

Energy Optimization Induces Predictive-coding Properties in a Multi-compartment Spiking Neural Network Model

PLOS Computational Biology

Dear Dr. Bohte,

Thank you for submitting your manuscript to PLOS Computational Biology. After careful consideration, we feel that it has merit but does not fully meet PLOS Computational Biology's publication criteria as it currently stands. Therefore, we invite you to submit a revised version of the manuscript that addresses the points raised during the review process.

Please submit your revised manuscript within 30 days Mar 30 2025 11:59PM. If you will need more time than this to complete your revisions, please reply to this message or contact the journal office at ploscompbiol@plos.org. Please include the following items when submitting your revised manuscript:

We look forward to receiving your revised manuscript.

Kind regards,

Emili Balaguer-Ballester, PhD

Academic Editor

PLOS Computational Biology

Daniele Marinazzo

Section Editor

PLOS Computational Biology

**Additional Editor Comments :**

Dear authors,

The reviewers appreciate your detailed work in addressing their questions. One of the reviewers made a minor comment on the tables and asked for some details about the resultant weight distribution. Thanks very much.

**Journal Requirements:**

At this stage, the following Authors/Authors require contributions: Mingfang Zhang, Raluca Chitic, and Sander Bohte. Please ensure that the full contributions of each author are acknowledged in the "Add/Edit/Remove Authors" section of our submission form.

**Reviewers' comments:**

Reviewer's Responses to Questions

Reviewer #1: The revised manuscript presents a computational model that explores how energy optimisation induces predictive coding properties in a multi-compartment spiking neural network. Whilst the authors have addressed previous comments and improved the detail of the model, there are some minor points that should be explored and clarified. By addressing these points, the authors can provide a more comprehensive and nuanced exploration of predictive coding in spiking neural network models.

The current analysis provides a foundation for understanding the learning dynamics of the network, but could benefit from additional insights into the structural properties of the network after learning. An examination of weight distributions and connectivity patterns could reveal nuanced computational mechanisms. In particular, an examination of the relative strengths of feedforward and feedback connections may reveal interesting parallels with existing neurophysiological research on neural information processing, such as Bruno and Sackman's research on thalamo-cortical connections. Second, the potential role of temporal precision and network performance provides a rich area for further investigation, for example by investigating the role of temporal jitter in the input. Last, the authors could consider exploring the spatial distribution of weights after learning, examining whether feedforward inputs exhibit the anatomically suggested precision and clustering (patchy characteristics), while feedback connections remain more diffuse.

minor points: Regarding the technical presentation, two minor points require attention. First, the tables (Tables 1 and 2) should include explicit units for all measurements to ensure clarity and reproducibility. Second, the term "inference" appears to be an inappropriate neologism and should be replaced with standard scientific terminology such as "inference" or "inferring".

Reviewer #2: The authors have addressed all my concerns.

**Have the authors made all data and (if applicable) computational code underlying the findings in their manuscript fully available?**

Reviewer #1: Yes

Reviewer #2: Yes

PLOS authors have the option to publish the peer review history of their article (what does this mean?). If published, this will include your full peer review and any attached files.

Reviewer #1: **Yes: **Laurent U Perrinet

Reviewer #2: **Yes: **Thomas Nowotny

**Figure resubmission:**
---

## [Decision Letter · Decision Letter 2]

3 May 2025

Dear Prof Dr. Bohte,

We are pleased to inform you that your manuscript 'Energy Optimization Induces Predictive-coding Properties in a Multi-compartment Spiking Neural Network Model' has been provisionally accepted for publication in PLOS Computational Biology.

Best regards,

Emili Balaguer-Ballester, PhD

Academic Editor

PLOS Computational Biology

Daniele Marinazzo

Section Editor

PLOS Computational Biology

The revised version is convincing for reviewers; they only have two (very) minor suggestions.

Reviewer's Responses to Questions

**Comments to the Authors:**

Reviewer #1: Thanks to the authors for their answer and congratulations for this nice paper.

**Have the authors made all data and (if applicable) computational code underlying the findings in their manuscript fully available?**

Reviewer #1: None

PLOS authors have the option to publish the peer review history of their article (what does this mean?). If published, this will include your full peer review and any attached files.

Reviewer #1: **Yes: **Laurent Udo Perrinet

---

## [Editor Report · Acceptance letter]

PCOMPBIOL-D-24-01385R2

Energy Optimization Induces Predictive-coding Properties in a Multi-compartment Spiking Neural Network Model

Dear Dr Bohte,

I am pleased to inform you that your manuscript has been formally accepted for publication in PLOS Computational Biology. Your manuscript is now with our production department and you will be notified of the publication date in due course.

With kind regards,

Zsofia Freund
